# Association between Fasting and Postprandial Levels of Liver Enzymes with Metabolic Syndrome and Suspected Prediabetes in Prepubertal Children

**DOI:** 10.3390/ijms24021090

**Published:** 2023-01-06

**Authors:** Katarzyna Bergmann, Anna Stefanska, Magdalena Krintus, Lukasz Szternel, Mauro Panteghini, Grazyna Sypniewska

**Affiliations:** 1Department of Laboratory Medicine, Collegium Medicum in Bydgoszcz, Nicolaus Copernicus University, 87-100 Torun, Poland; 2Research Centre for Metrological Traceability in Laboratory Medicine (CIRME), Department of Biomedical and Clinical Sciences, University of Milan, 20157 Milan, Italy

**Keywords:** metabolic syndrome, liver enzymes, prediabetes

## Abstract

Elevated liver enzyme activity may be associated with metabolic syndrome (MetS); however, it is not included in the MetS definition for children. Postprandial changes in the levels of biochemistry tests are related to manifestations of metabolic abnormalities. We assessed the association between fasting and postprandial liver enzymes levels with MetS and elevated hemoglobin A1c (HbA1c) in children aged 9–11. The study included 51 girls and 48 boys, all presumably healthy. In all participants’ anthropometric indices, fasting glucose, insulin, lipid profile and HbA1c were measured. Enzymes, including alanine aminotransferase (ALT) and gamma-glutamyl transferase (GGT), were assayed in fasting and postprandial states. Individuals were divided into subgroups: with (MetS(+): n = 26); without MetS (MetS(−): n = 73); with HbA1c levels ≤ 5.3% (n = 39); and ≥5.7% (n = 11). Elevated fasting GGT levels were found in 23% of MetS(+) children and rarely in MetS(−) children; increased postprandial GGT was noted in 35% of MetS(+) individuals. Postprandial GGT changes tend to predict MetS (OR = 1.16; *p* = 0.092). Increased fasting ALT was found rarely in MetS(+) children, but did not occur in MetS(−) children. HbA1c ≥ 5.7% occurred rarely and neither fasting ALT nor GGT were related to elevated HbA1c. However, postprandial change of ALT was a good positive predictor of increased HbA1c (OR = 1.33; *p* = 0.021). Postprandial GGT performs better as an indicator of metabolic syndrome occurrence, and instead postprandial ALT may predict prediabetes in prepubertal children.

## 1. Introduction

The growing prevalence of metabolic syndrome (MetS), a cluster of metabolic abnormalities associated with obesity, is a global health problem identified from childhood into adulthood. The individual components of MetS may occur together, driven by mechanisms leading to insulin resistance and linked to inflammatory processes. MetS is linked to obesity-related diseases, including non-alcoholic fatty liver disease, and increases the risk of developing cardiovascular diseases and diabetes type 2 in children [1,2,3]. Early diagnosis of MetS, particularly in obese children, is of utmost importance as the prevalence of obesity-associated health consequences is continuously growing. Diagnosis of MetS, based on clinical parameters defined according to the recommendations for children and adolescents, does not include assessment of liver function enzymes [1,4]. Notwithstanding, in children, even moderately elevated aminotransferase activity was shown to be associated with disturbances in the metabolism of lipids and glucose [5,6].

It is assumed that the assessment of routine clinical chemistry analytes should be performed in the conditions of normal daily activity, since people do not fast for about 16 h a day. Postprandial change in the levels of analytes in the blood, in general, does not significantly affect the assessment of results as compared with the same test in the fasting state, but may reflect disturbances in metabolism [7]. The introduction of non-fasting testing for selected biochemical parameters, apart from simplifying the preanalytical phase, would also save children from the discomfort associated with the blood sampling procedure, in particular, the need for food restrictions, which can be difficult to perform and generate additional stress related to blood sampling, as well as reluctance to undergo further tests. In this study, we aimed to assess the association between fasting and postprandial levels of liver function enzymes with the occurrence of metabolic syndrome and elevated hemoglobin A1c level in a group of presumably healthy prepubertal school children aged 9–11.

## 2. Results

The characteristics of study participants, dependent on the presence of MetS and elevated HbA1c, are displayed in Table 1. All data in this Table, except delta values for ALT and GGT (a difference between fasting and postprandial enzyme activity), represent the results of measurements performed in the fasting state. Overall, in our study group of apparently healthy children, there were 51 girls and 48 boys. Out of all children, 10 girls and 16 boys were identified as having MetS. Children with MetS (MetS(+)) had significantly higher median values of measured enzymes, CRP and assessed indices, except for HbA1c which was similar regardless of the occurrence of MetS. Fasting GGT level over the upper reference limit (URL > 21 IU) was found mostly in children with MetS(+) than in these without (23% vs. 4%). Postprandial GGT levels over the URL were noted much more frequently in MetS(+) individuals than in MetS(−) (35% vs. 5%; *p* < 0.001). Of note, fasting ALT levels over the URL occurred only in 8% of MetS(+) children, but did not occur in these MetS(−), just like elevated postprandial ALT levels were not observed in both groups.

In the whole study group, 39 children (39.4%) presented with desirable HbA1c levels ≤5.3%, whereas only 11 (11.1%) had elevated HbA1c (≥5.7%; ≥39 mmol/mol). Nevertheless, elevated HbA1c was found more frequently in MetS(−) children than among MetS(+) (12.3 vs. 7.7%), similarly to HbA1c ≤5.3% (41.1% vs. 34.6%). Comparison of the two groups, based on HbA1c concentration, showed that only HOMA-IR and delta ALT were significantly higher in children with elevated HbA1c.

The analyses of Spearman correlations in the whole study group indicated a positive relationship between TG concentrations and TG/HDL-C and the activity of ALT (r = 0.452 and r = 0.489; *p* < 0.0001) and relationship with GGT activity (r = 0.265 and r = 0.248; *p* = 0.010 and *p* = 0.016, respectively). Moreover, both ALT and GGT were significantly, although weakly, related to CRP (r = 0.268; *p* = 0.007 and r = 0.261; *p* = 0.009).

Logistic regression analysis was performed to assess the relationships between the activity of enzymes measured in the fasting state, calculated indices and the presence of MetS. The models in the univariable analysis indicated that WTI and TG/HDL-C were the most significant predictors of having MetS (Table 2). The WTI value explained 75% of the variation in the presence of MetS and had excellent discriminatory power. It has to be stressed that the activities of ALT and GGT, which are not considered as components for the definition of MetS, were shown to be significant positive predictors for the presence of MetS as well. The likelihood of MetS occurrence was 29% per one unit increment of ALT, and 16% per one unit increase in GGT.

Based on the result of Nagelkerke R2, the models with ALT and GGT explained 36% and 23% of the variation for the presence of MetS. The ROC curve analysis showed that ALT had a very good diagnostic accuracy for prediction of the occurrence of MetS, whereas the diagnostic accuracy of GGT was lower. The other model designed for the prediction of elevated HbA1c occurrence showed that HOMA-IR was a good significant positive predictor [OR = 1.71(95% CI 1.04–2.81); *p* = 0.035].

In order to assess which of the measured liver enzymes are most strongly associated with the presence of MetS, we performed a multivariable logistic regression (Table 3). The analysis revealed that fasting ALT was a better positive predictor of MetS than GGT. The designed model explained 44% of the variation for the presence of MetS and showed a very good diagnostic accuracy.

Furthermore, we performed a more in-depth analysis to demonstrate whether postprandial changes in the activity of aminotransferases are associated with the presence of MetS or elevated HbA1c. The analysis of postprandial changes (at 2–4 h after habitual breakfast) was performed based on the comparison of ALT deltas or GGT deltas, calculated as mean values of individual differences between fasting and non-fasting enzyme activity (Table 1). As indicated in Table 1, the mean delta values for GGT and ALT did not significantly differ between the groups with and without MetS, or with elevated and desirable HbA1c levels, except the delta for ALT which was significantly higher in children with elevated HbA1c. Of note, in the whole study group, delta changes for ALT correlated weakly but significantly with HbA1c (r = 0.300; *p* = 0.002), but no relationship was found for delta GGT. 

Next, the analysis of deltas for ALT and GGT was performed in subgroups depending on the occurrence of MetS and/or elevated HbA1c and was based on the comparison of RCV (Figure 1). The frequency of deltas above or below the RCV cut-offs was calculated (RCV cut-off for ALT was accepted as 47.4% and RCV cut-off for GGT as 8%). This analysis revealed that the frequencies of GGT delta changes exceeding the RCV cut-off were similar, irrespective of the presence of MetS. It is worth noting that deltas of GGT over the RCV occurred two-fold more frequently in MetS(+) subjects than deltas of ALT (46% vs. 19%). This could suggest that major postprandial GGT, but not ALT, changes may be indicative of metabolic disturbances associated with MetS. 

The frequency of delta changes of ALT was similar in MetS(−) subjects. On the other hand, we observed that the frequency of delta ALT over the RCV cut-off was much lower in MetS(+) subjects than the frequency of delta ALT below the RCV (19% vs. 81%, Figure 1A). To analyze this phenomenon, we checked whether there was an association between fasting and postprandial ALT activity, presented as delta above the RCV cut-off. We found that individuals with delta over the RCV had a significantly lower activity of fasting ALT, than those with delta below the RCV [3.0 IU/L (2.0–6.0) vs. 11.0 IU/L (6.0–14.0); *p* < 0.001]. However, low values of fasting ALT, observed mostly in children without MetS (Table 1), were found very rarely in MetS(+). Low fasting ALT were more frequently related to greater changes in postprandial state, which could also be explained by the negative association between delta ALT and MetS occurrence (Table 4). 

The presence of elevated HbA1c was associated, most of all, with a higher frequency of ALT delta changes over the RCV, when compared to subjects with desirable HbA1c (55% vs. 28%; *p* = 0.06). The highest frequency of ALT deltas below RCV was found in subjects with a desirable HbA1c, compared to those with elevated HbA1c (72 vs. 45%; *p* = 0.013). The frequencies of deltas over the RCV cut-off for GGT were similar, regardless of HbA1c level.

Univariable logistic regression analysis disclosed that postprandial change of GGT, presented as delta, showed a tendency to predict the presence of MetS, but its diagnostic accuracy was poor (Table 4) and disappeared after inclusion of GGT RCV cut-off in the frequency analysis of deltas (Figure 1). On the other hand, postprandial change of ALT was negatively related with MetS presence, but was a good significant positive predictor of elevated HbA1c, which explained almost 22% of the variation and had a good discriminatory power (Table 4). The association between the HbA1c ≥ 5.7% occurrence and delta ALT was independent of fasting ALT activity.

## 3. Discussion

The prevalence of metabolic syndrome in children depends on age, sex, ethnicity and applied criteria. Similarly, the prevalence of prediabetes in children depends on the method used for the assessment of prediabetes phenotype. Among European children, the prevalence of MetS is high and amounts up to 16%, and that of prediabetes, diagnosed on the basis of HbA1c, is similar (16.3%) [8,9].

There are several definitions used for diagnosis of MetS in the pediatric population and in a recently published review, the discrepancies between definitions have been discussed [8]. Considering the high prevalence of obesity and hypertriglyceridemia among the pediatric population in our country, we applied the IDF criteria from 2007 to define MetS which are quite comparable with those published earlier by Ford et al. [1,10,11].

Diagnosis of MetS, according to the recommendations for children and adolescents, does not include the assessment of liver enzymes [1,12].

Studies in children on the possible association between liver-function enzymes and obesity-related diseases, such as metabolic syndrome and NAFLD, are limited but indicate the significant relationship between the activity of ALT and MetS components [13,14]. In obese Japanese children with MetS or type 2 diabetes (T2DM), ALT showed a very good discriminatory power for the onset of T2DM [14]. Another recent large European multicenter cross-sectional study which assessed the prevalence of prediabetes and T2DM in obese children, with varying degrees of elevated transaminases, showed much higher risk of prediabetes and T2DM if the increase in ALT activity was up to two-fold the upper reference limit [6].

The levels of GGT, for a long time, were related to the onset of metabolic syndrome and were linked to an increased risk of T2DM [15,16]. GGT, however, may be also a predictor of cardiovascular diseases and heart failure, and an elevated level of GGT is a consequence of non-alcoholic fatty liver disease (NAFLD) [17]. 

In healthy children and adults, most blood biochemistry results are not affected by the non-fasting state [7,18]. Pediatric age and sex-specific reference intervals reported for 40 common biochemical markers have been established on the non-fasting blood samples from over 9700 children 19]. In a subset of children, the levels of routine analytes were measured at fasting, postprandial and random time points and it was shown that most of them, except triglycerides and glucose, did not considerably change, and therefore can be measured anytime 19]. Obtaining fasting samples from the pediatric population is challenging, whereas taking non-fasting samples creates undeniable convenience and, most of all, leads to increased participation in any screening study. What is important, postprandial changes in the serum levels of routine biochemistry tests may be related to early manifestations of metabolic abnormalities, particularly in subjects with dyslipidemia and insulin resistance [7,19]. It is worth noting one previous study on prepubertal obese children with metabolic syndrome, which reported that cardiometabolic risk was associated with increased postprandial, but not with fasting, insulin resistance [20]. To the best of our knowledge, no studies, so far, assessed the association between fasting and postprandial levels of liver-function enzymes with the occurrence of metabolic syndrome and elevated hemoglobin A1c in prepubertal children. 

The comparison of postprandial changes of ALT and GGT levels, based on the analysis of deltas, over or below the reference change values (RCV), demonstrated that delta changes of GGT, which exceeded the RCV, occurred two-fold more frequently in subjects with MetS than delta changes of ALT (46% vs. 19%). We suggest that in MetS(+) individuals, GGT, rather than ALT, turns out to be more sensitive to postprandial changes and that major changes of GGT may be indicative of metabolic disturbances associated with MetS (Table 1, Figure 1). This suggestion was confirmed by regression analysis which disclosed that postprandial change of GGT shows a tendency to predict the presence of MetS, whereas the relationship between delta ALT and MetS occurrence is meaningless (Table 4). It should be highlighted that fasting GGT levels over the URL were found in every fourth child with MetS and very rarely in MetS(−), but elevated postprandial GGT was noted in every third MetS(+) individual, compared to MetS(−) (*p* < 0.001). Moreover, mean delta change of GGT, expressed in units of activity, was almost three-fold (Table 1), though not significantly, higher in children with MetS than in MetS(−). All these findings presented above could indicate that GGT in children with MetS is more sensitive to the effects of a meal, which does not pertain to MetS(−) individuals.

It is worth noting that fasting ALT levels over the URL were found only in 8% of children MetS(+), but did not occur in the MetS(−) children, just like elevated postprandial ALT levels were not observed either in MetS(+) or in MetS(−). In summary, this shows that ALT levels are not significantly affected by a meal and explains why ALT assessed in the fasting state seems to be a better predictor of MetS than fasting GGT (OR = 1.2 vs. 1.09; AUC = 0.82 vs. 0.76).

In our study, participants with MetS were characterized by a significantly higher TG/HDL-C which was the second best predictor of MetS. In contrast, fasting glucose and HbA1c levels were similar, irrespective of MetS occurrence, as indicated in our previous study on children [21]. The correlation between GGT activity and lipid biomarkers (TG and TG/HDL-C), the significant increase in these two biomarkers towards higher GGT activity in tertiles (*p* = 0.018), but most importantly the association of postprandial changes with the presence of MetS, allow us to suggest that GGT level may indicate metabolic disturbances associated with lipid metabolism in children with MetS.

The associations between liver enzymes and factors related to MetS were assessed earlier in prepubertal children with and without obesity. Children with obesity had significantly elevated ALT levels. In addition, in obese children, ALT correlated significantly with HOMA-IR, CRP and TG [13]. Our overall study group included, solely, 23% obese subjects, but we found similar, though weaker, correlations between both ALT, GGT and TG and CRP.

Another recent study performed on healthy children 5–12 years reported on TG/HDL and uric acid (UA), also considered as factors predisposing to metabolic disorders which are not included as components in the definition of MetS [22]. ALT and GGT levels were significantly higher in children with a predisposition for MetS compared to those without predisposition to MetS, whereas UA and TG/HDL were similar in both groups. In subjects with a predisposition to MetS, TG/HDL-C significantly correlated with GGT, but not with ALT, whereas no correlation was observed between UA and both enzymes. The authors highlighted the contributing role of TG/HDL-C in the development of MetS in children, indicating hepatic dysfunction and predisposition to cardiovascular complications. This is in line with the findings of the current study which indicated that TG/HDL-C was one of the best predictors of MetS (OR = 8.67; AUC = 0.88).

Elevated HbA1c ≥ 5.7% occurred rarely in our study group (11%), and neither fasting ALT nor GGT levels were related to elevated HbA1c. It should, however, be mentioned that delta ALT was significantly higher in children with an elevated HbA1c and, what is more important, elevated HbA1c was associated with a higher frequency of ALT delta changes over the RCV. Postprandial changes of ALT were a good positive predictor of elevated HbA1c. Noteworthy was that the association between HbA1c ≥ 5.7% occurrence and delta ALT was independent of fasting ALT activity. Therefore, we imply that postprandial ALT levels may be linked to the risk of prediabetes in prepubertal children. The biological explanation for the finding that postprandial change of ALT was a good positive predictor of increased HbA1c is difficult because this was not reported by other authors. We can only hypothesize that postprandial increase in ALT activity might be associated with liver fat accumulation accompanied by the effects of glucose toxicity on the liver, which are observed during prolonged hyperglycemia in MetS [23]. Moreover, the liver is susceptible to the effects of hyperglycemia-induced oxidative stress, which may lead to liver tissue injury [24]. Another possible explanation is that the oxidative stress, accompanied by elevated glucose level, promotes the formation of advanced glycation end-products (AGEs), including HbA1c formation [25,26].

The results of our study should be interpreted in light of some limitations. We were unable to collect detailed dietary reports; however, the study excluded underweight children.

Based on our results, we cannot clearly exclude or confirm the presence of NAFLD in the subjects. Elevated liver enzymes, in particular ALT, exceeding two-fold the upper limit of normal values may be used as surrogate for “mild NAFLD” [6,27]. An earlier study performed in the Polish pediatric population showed that in obese children, with ultrasonography confirmed NAFLD, fasting levels of ALT were in the range of 51–95 U/L and those of GGT were in the range of 22–45 U/L [28]. On one hand, occurrence of NAFLD in our MetS(+) group seems to be unlikely because the single highest level of fasting ALT was 27 U/L at TG level of 188 mg/dL, and the single highest fasting GGT level was 38 U/L at TG 111 mg/dL. On the other hand, it is worth noting that 54% of MetS(+) subjects had elevated fasting TG (≥110 mg/dL) and in four cases, TG level was in the range of 152–277 mg/dL. Several studies confirmed that hypertriglyceridemia is associated with NAFLD and among other lipid parameters, it might be considered as the strongest predictor of NAFLD [29,30]. Moreover, a recent cross-sectional study performed among 3170 10-year-old potentially healthy children showed that the NAFLD prevalence was 9.1% and 25.0% among children who were overweight and obese, respectively. The authors also observed that both higher liver fat and NAFLD were most strongly associated with higher values of systolic blood pressure, insulin resistance, triglycerides and CRP [31]. Therefore, we hypothesize that the relationship between changes in liver enzyme activity and MetS and HbA1c levels might be associated with hepatocyte disorders due to early stage steatosis; however, this requires further investigation using imaging methods.

As another limitation of the present study, we have to mention the small sample size, in particular the very small numbers of children with metabolic syndrome; thus, the generalizability of our results is uncertain. On the other hand, the current study included quite a homogenous group of children, in terms of the narrow age range and prepubertal stage (90% of children without MetS reported Tanner stage 1 (TS1) and in a group with MetS 70% reported TS1). Blood samples were collected in fasting and non-fasting states from the same individuals and enzyme measurements were performed in duplicates from each individual in the same run.

## 4. Conclusions

In conclusion, the presented data allowed us to suggest that postprandial GGT performs better as an indicator of metabolic disturbances in children with MetS, and instead postprandial ALT is a predictor of prediabetes in prepubertal children. We advocate for the introduction of non-fasting testing for selected biochemical parameters which would be of significant benefit, saving children from the discomfort of being fasted the while blood sampling procedure occurs.

## 5. Materials and Methods

### 5.1. Subjects

The study consisted of 99 children (51 girls, 48 boys) aged 9–11 from four primary schools in one suburban region. Subjects were randomly selected from a cohort of 284 presumably healthy school children aged 9–11, enrolled in a previous cross-sectional study [8]. In the study group, 73 subjects (41 girls, 32 boys) had no features of metabolic syndrome (MetS), while 26 children (10 girls and 16 boys) presented central obesity and at least two other features of MetS (fasting hyperglycemia, fasting dyslipidemia and/or increased blood pressure).

### 5.2. Methods

Fasting and postprandial venous blood was collected in the morning in order to obtain serum and whole blood samples. The fasting blood sample was collected after at least 8 h of fasting, whereas the second sample was taken after 2–4 h since the last meal (habitual breakfast); the time interval between the two samplings was not longer than 14 days [32].

In all participants, the following laboratory tests were determined: glucose, insulin, triglycerides (TG), HDL-cholesterol (HDL-C), C-reactive protein (CRP), glycated hemoglobin (HbA1c) were measured immediately after collection. Alanine aminotransferase (ALT), aspartate aminotransferase (AST), gamma-glutamyl transferase (GGT) and alkaline phosphatase (ALP) were assayed in fasting and non-fasting previously deep frozen serum samples, in duplicates from each individual in the same run. Each sample underwent, in advance, an assessment of hemolysis index to exclude an interference on ALT, AST and ALP activity. All laboratory tests were performed at the Department of Laboratory Medicine, Collegium Medicum, Nicolaus Copernicus University in Torun on the Horiba ABX Pentra 400 (Horiba Medical, Montpellier, France), except for measurements of enzymes which were performed on the Alinity c platform (Abbott Laboratories, Chicago, IL, USA) and HbA1c which was determined with the use of Bio-Rad D10 (Bio-Rad Laboratories, Hercules, CA, USA).

Anthropometric measurements were performed, and blood pressure was measured by a qualified nursing team on the same day that blood samples were taken. In all subjects, waist circumference measurements were obtained, and waist/triglycerides indexes (WTI) were calculated using the following equation: ln [TG(mg/dL)·WC(cm)/2] [33]. Body mass index (BMI) and BMI percentiles were calculated with the use of an online calculator for children and adolescents in Poland, as previously described [8]. On the basis of glucose and insulin concentrations, the homeostasis model assessment (HOMA-IR) was calculated. For the purpose of statistical analysis, 50 children were included in subgroups according to HbA1c levels: ≤5.3% (n = 39) and ≥5.7% (n = 11) [34]. 

Parental written consent was obtained for each participant. The study was approved by the Bioethics Committee at Nicolaus Copernicus University Collegium Medicum in Bydgoszcz, Poland (no. KB 338/2015, annex 487/2019) and carried out in accordance with the Declaration of Helsinki.

BMI percentile classification was applied according to the guidelines of the International Obesity Task Force: underweight BMI <5; overweight ≥85 and <95; obesity ≥95 percentile [35]. Participants were divided into two groups: without MetS—presenting without any component of metabolic syndrome, and MetS group—presenting with central obesity and at least 2 other components of MetS: central (abdominal) obesity defined by a waist circumference ≥90th percentile by sex and age for European population; fasting blood glucose ≥ 100 mg/dL; increased systolic blood pressure (SBP) or diastolic blood pressure (DBP) ≥ 90th percentile by sex, height and age; fasting dyslipidemia defined as TG ≥110 mg/dL and low HDL-C as <40 mg/dL [1,4]. 

Pediatric reference intervals for age and sex, related to the instrumentation and applied methodology, for ALT 9–25 IU/L and GGT 6–21 IU/L were accepted according to Adeli et al. [19]. For the purpose of comparison of the delta change (a difference between enzyme activity in fasting and postprandial samples), a special tool, called reference change value (RCV), was applied. Biological variation is important when interpreting laboratory test results providing information on within-subject physiological changes; however, RCV is a suitable tool for comparison which provides information on both the within-subject biological variation and analytical variation [36]. On the basis of data established for the pediatric population, the value of RCV for ALT was accepted as 47.4% and 8% for GGT [36].

### 5.3. Statistical Analysis

Statistical analysis was performed using Statistica 13.3 (StatSoft Inc., Tulsa, OK, USA, 2019) and MedCalc 19.1.7 (MedCalc Software Ltd., Ostend, Flanders, 2020) software. The data were presented as medians and 25th and 75th percentiles (non-Gaussian distribution) or mean ± standard deviation (SD). The variables were compared using the Mann–Whitney U test. Parameters with non-Gaussian distribution were normalized by natural log transformation for univariate logistic regression. In all logistic models, odds ratios (ORs) were calculated for a 1 standard deviation (SD) increase in independent variables. The level of statistical significance (*p*) was set as 0.05.

## Figures and Tables

**Figure 1 ijms-24-01090-f001:**
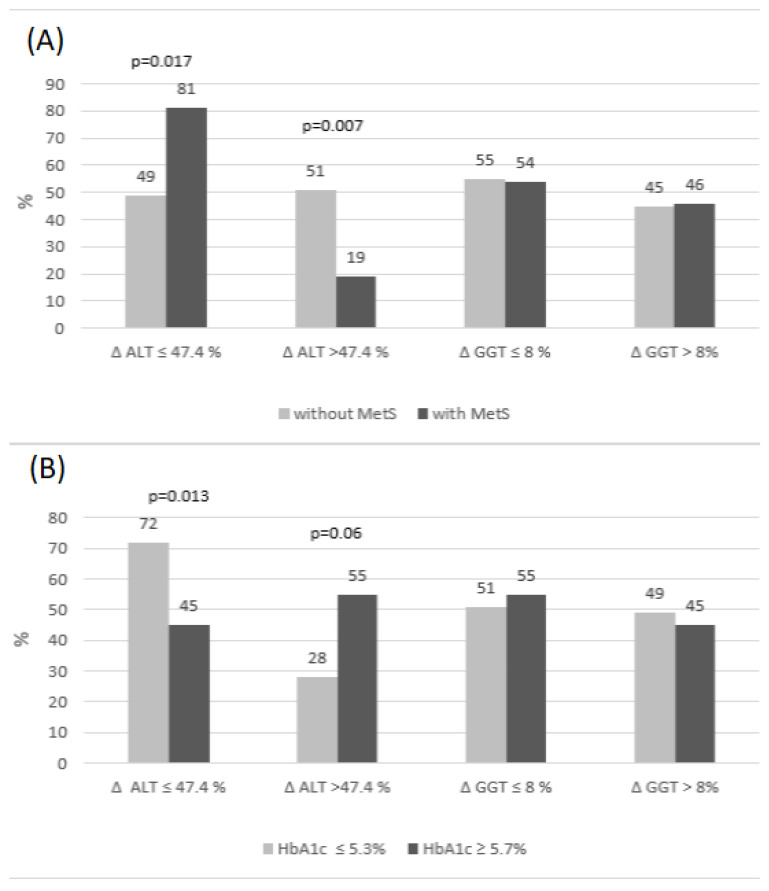
The percentage of deltas below and above RCV for GGT and ALT in the subgroups with/without MetS (**A**) and elevated/desirable HbA1c (**B**).

**Table 1 ijms-24-01090-t001:** Basal characteristics of study groups.

Variables	MetS(−) (n = 73)	MetS(+) (n = 26)	*p*	HbA1c ≤ 5.3% (n = 39)	HbA1c ≥ 5.7 (n = 11)	*p*
Age (years)	10 (9–10)	10 (9–10)	0.676	10 (9–10)	10 (9–10)	0.669
Sex F/M (F%)	40/33 (55)	10/16 (38)	0.137	17/22 (44)	6/5 (55)	0.518
BMI centile	42 (24–63)	96 (95–97)	<0.001	45 (24–86)	43 (9.0–78)	0.511
HOMA-IR	1.58 (1.07–2.38)	3.13 (2.47–4.91)	<0.001	1.58 (1.14–2.50)	2.70 (2.14–3.45)	0.008
TG/HDL-C	0.89 (0.66–1.50)	2.44 (1.63–3.03)	<0.001	1.16 (0.73–1.87)	1.38 (0.65–1.93)	0.854
GGT (IU/L)	11 (9–14)	17 (12–21)	<0.001	11 (9–15)	12 (10–16)	0.689
ALT (IU/L)	5 (3–8)	12 (9–16)	<0.001	8 (5–16)	6 (4–8)	0.300
AST (IU/L)	29 (27–31)	32 (29–38)	0.006	30 (28–33)	28 (25–32)	0.133
ALP (IU/L)	168 (140–229)	216 (167–240)	0.019	180 (140–225)	166 (113–231)	0.699
WTI	7.51 (7.24–7.88)	8.47 (8.32–8.71)	<0.001	7.77 (7.26–8.11)	7.89 (7.26–8.15)	0.892
HbA1c (%)	5.4 (5.2–5.5)	5.4 (5.3–5.5)	0.522	5.2 (5.1–5.3)	5.7 (5.7–5.8)	<0.001
CRP (mg/L)	0.46 (0.14–1.10)	1.82 (0.75–3.03)	<0.001	0.65 (0.14–1.49)	0.34 (0.15–1.86)	0.725
Δ ALT U/L	1.7 ± 4.6	1.10 ± 5.19	0.590	−0.65 ± 4.2	2.95 ± 3.14	0.011
Δ GGT U/L	0.40 ± 3.4	1.13 ± 4.41	0.359	−0.06 ± 3.2	0.86 ± 3.11	0.39

Results are presented as median (Q1–Q3) or mean ± standard deviation (SD); MetS, metabolic syndrome; BMI, body mass index; HOMA-IR, homeostatic model assessment—insulin resistance; TG/HDL-C, triglyceride to high-density lipoprotein cholesterol ratio; GGT, gamma-glutamyl transferase; ALT, alanine aminotransferase; AST, aspartate aminotransferase; ALP, alkaline phosphatase; WTI, waist-triglyceride index; HbA1c, glycated hemoglobin. F-females; M-males; Δ (delta) is defined as a difference between fasting and postprandial activity of liver enzymes.

**Table 2 ijms-24-01090-t002:** Univariable logistic regression analysis for the prediction of MetS in the study group.

Variable	Study Group (n = 99)			
	*p*	OR (95% CI) per 1 Unit	NR2	AUC (95% CI)
HOMA-IR	<0.001	1.81 (1.33–2.46)	0.27	0.81 (0.72–0.88)
TG/HDL	<0.001	8.67 (3.41–22.06)	0.52	0.88 (0.80–0.94)
ALP	0.122	1.0 (0.99–1.01)	0.04	0.64 (0.53–0.73)
GGT	<0.001	1.16 (1.06–1.26)	0.23	0.76 (0.67–0.84)
ALT	<0.001	1.29 (1.15–1.44)	0.36	0.82 (0.73–0.89)
AST	0.530	1.01 (0.97–1.05)	0.01	0.67 (0.57–0.76)
WTI	<0.001	14.50 (5.20–8.12)	0.75	0.96 (0.89–0.99)

HOMA-IR, homeostatic model assessment—insulin resistance; TG/HDL-C, triglyceride to high-density lipoprotein cholesterol ratio; ALP, alkaline phosphatase; GGT, gamma-glutamyl transferase; ALT, alanine aminotransferase; AST, aspartate aminotransferase; WTI, waist–triglyceride index; NR2, Nagelkerke R^2^; OR, odds ratio; CI, confidence interval; AUC, area under the curve.

**Table 3 ijms-24-01090-t003:** Multivariable regression analysis for MetS prediction by fasting levels of liver enzymes.

Variables	OR (95% CI) per 1 Unit (n = 99)	*p*
GGT	1.09 (1.00–1.19)	0.046
ALT	1.25 (1.11–1.42)	<0.001
ALP	1.01 (0.99–1.02)	0.157

Nagelkerke R^2^ of the model = 0.44; Cox & Snell R^2^ = 0.30; AUC = 0.86 (0.77–0.92). GGT, gamma-glutamyl transferase; ALT, alanine aminotransferase; ALP, alkaline phosphatase; OR, odds ratio; CI, confidence interval.

**Table 4 ijms-24-01090-t004:** Univariable logistic regression analysis for MetS and HbA1c ≥ 5.7% prediction by postprandial changes in the activity of aminotransferases.

MetS(−) vs. MetS(+)				
Variable	*p*	OR (95% CI) per 1 Unit	NR2	AUC (95% CI)
Δ GGT	0.092	1.16 (0.97–1.39)	0.040	0.58 (0.47–0.68)
Δ ALT	0.588	0.97 (0.88–1.07)	0.004	0.54 (0.44–0.64)
**HbA1c ≤ 5.3% vs. ≥5.7%**				
**Variable**	** *p* **	**OR per 1 Unit (95% CI)**	**NR2**	**AUC (95% CI)**
Δ GGT	0.384	1.10 (0.88–1.38)	0.024	0.61 (0.46–0.74)
Δ ALT *	0.021	1.33 (1.04–1.69)	0.216	0.74 (0.59–0.85)

* The association is still significant after adjustment for fasting ALT activity (*p* = 0.038); Δ (delta) is defined as a difference between fasting and postprandial activity of liver enzymes. MetS, metabolic syndrome; HbA1c, glycated hemoglobin; GGT, gamma-glutamyl transferase; ALT, alanine aminotransferase; NR2, Nagelkerke R^2^; OR, odds ratio; CI, confidence interval; AUC, area under the curve.

## Data Availability

Not applicable.

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
