# Peer review of "Association between Fasting and Postprandial Levels of Liver Enzymes with Metabolic Syndrome and Suspected Prediabetes in Prepubertal Children"

_ijms, 2023, doi:10.3390/ijms24021090_

Round 1

Reviewer 1 Report

Nice study in children with and without MS;  however , there are much more defintions of MS in childhood published;  the authors shouls state, why they selected this form of definition. There is no conclusion.

Author Response

Thank you very much for this important comment. On page 7, we explained that we used the current IDF guidelines for the diagnosis of MetS, due to the relatively high prevalence of obesity and hypertriglyceridemia in pediatric population in our country. The conclusions were presented in the last paragraph of the discussion, however we decided to extract this paragraph, according to your suggestion.

Reviewer 2 Report

This is a small study of a cohort of children with and without the metabolic syndrome. The aim is to evaluate the association between fasting and post-prandial levels of liver function enzymes with occurrence of metabolic syndrome and elevated haemoglobin A1c values. The Authors conclude that postprandial GGT performs better as an indicator of metabolic disturbances, instead postprandial ALT may predict prediabetes in prepubertal children.

However, some aspects can cause interpretation problems and indicate conclusions that are difficult to extend to the large number of children with the metabolic syndrome. There are some issues that need to be addressed/clarified to ensure that the data fully support the statements and conclusions made.

I suggest that the Authors give some comments on the following points:

·       Elevated liver enzymes activity is not included in MetS definition both in children and in adults, but the Authors say that is associated to MetS.

·       Liver enzyme activity may be elevated as the consequence of the presence of an underlying fatty liver disease (which was not assessed in the present study), which is considered as the hepatic component of MetS.

·       Fasting elevation of liver enzymes was observed only in a minority of children with MetS.

·       Postprandial liver enzyme testing is not a well standardized diagnostic method.

·       How to explain that postprandial change of ALT was a good positive predictor of increased HbA1c?

·       Why the introduction of non-fasting testing for selected biochemical parameters, apart from simplifying the preanalytical phase, would save children from the discomfort associated with the blood sampling procedure?

·       Aims, conclusions and take-home messages of the study are unclear

Author Response

I suggest that the Authors give some comments on the following points:

  • Elevated liver enzymes activity is not included in MetS definition both in children and in adults, but the Authors say that is associated to MetS.

Thank you very much for this comment. In the Introduction, at the beginning of page 2, it was emphasized that the MetS criteria do not take into account the activity of liver enzymes. However, we suggested the link between elevated enzymes and disorders of carbohydrate and lipid metabolism, which was observed in several studies (References 5 and 6).

  • Liver enzyme activity may be elevated as the consequence of the presence of an underlying fatty liver disease (which was not assessed in the present study), which is considered as the hepatic component of MetS.

Thank you very much for this important comment. We totally agree that metabolic disorders in MetS might be associated with liver steatosis. We added explanation of potential relationship and mechanisms on pages 7-9 and included new references (References 18-23 and 28-33). We also presented the lack of imaging studies as one of the limitations of the study (page 9).

  • Fasting elevation of liver enzymes was observed only in a minority of children with MetS.

Thank you very much for this comment. On page 9 we explain that diagnosis of NAFLD seems to be unlikely in our MetS(+) group, based on enzymes levels. However, hypetriglyceridemia was found in 54% of  MetS(+) group and according to several studies, it might be considered as the strong predictor of NAFLD. We included additional references (References 34, 35, 36). Therefore, we hypothesize that the relationship between changes in liver enzymes activity and MetS and HbA1c levels might be associated with hepatocyte disorders due to early stage steatosis, however this requires further investigation, using imaging methods.

  • Postprandial liver enzyme testing is not a well standardized diagnostic method.

Thank you very much for this comment. On page 7, we have described studies confirming that fasting is not required for most biochemical tests also in pediatric populations (References 7, 12, 24). We would like to mention here, the “revolutionary” paper published in Eur Heart J 2016;37:1944-1958  “Fasting is not routinely required for determination of a lipid profile”  which presented data for large sample populations of adults and children as well.

  • How to explain that postprandial change of ALT was a good positive predictor of increased HbA1c?

Thank you very much for this comment. On page 8, we have described potential mechanisms linking postprandial ALT with changes in HbA1c.

  • Why the introduction of non-fasting testing for selected biochemical parameters, apart from simplifying the preanalytical phase, would save children from the discomfort associated with the blood sampling procedure?

Thank you very much for this comment. In the Introduction (page 2) we explained that the discomfort  of being fasted for laboratory tests is related to the need for food restrictions and long breaks between meals, which can be difficult to perform (especially in younger children) and generates additional stress related to blood sampling, as well as reluctance to undergo further tests.

  • Aims, conclusions and take-home messages of the study are unclear

High prevalence of MetS in pediatric population and the strong association with future diabetes and CVD is a great cause of concern. Considering the high prevalence of overweight/obesity (20%), dyslipidemia/ hypertriglyceridemia (over 30%) and fasting hyperglycemia (approx. 17%)  which come together with MetS in the general population of children in our country we intended to elucidate the relationships between liver-function enzymes and  MetS as well as suspected prediabetes.

Having in mind earlier reports which showed that  elevated liver enzymes are associated with abnormalities in lipid and glucose metabolism and being aware that postprandial levels of routine biochemistry tests may manifest early metabolic disturbances we aimed to assess the possible associations between selected liver enzymes and MetS and suspected prediabetes in a cohort of prepubertal children. 

Non-fasting testing, particularly in children and adolescents, is widely applied in several European countries, Canada and US and there are recommendations  to perform non-fasting routine lipid profile for adults and children; therefore in the absence of formal standardized screening procedures for MetS itself we aimed to add some new information on this topic.

We slightly modified the Abstract, Introduction (pages 1-2) and Conclusions to make them more clear and convincing.

Round 2

Reviewer 1 Report

THe authors made adequate ammendmennts

Reviewer 2 Report

The Authors replied all questions.